# ER+ Breast Cancer Strongly Depends on MCL-1 and BCL-xL Anti-Apoptotic Proteins

**DOI:** 10.3390/cells10071659

**Published:** 2021-07-02

**Authors:** Clara Alcon, Jorge Gómez Tejeda Zañudo, Reka Albert, Nikhil Wagle, Maurizio Scaltriti, Anthony Letai, Josep Samitier, Joan Montero

**Affiliations:** 1Institute for Bioengineering of Catalonia (IBEC), The Barcelona Institute of Science and Technology (BIST), 08028 Barcelona, Spain; calcon@ibecbarcelona.eu (C.A.); jsamitier@ibecbarcelona.eu (J.S.); 2Eli and Edythe L. Broad Institute of MIT and Harvard, Cambridge, MA 02142, USA; jgtz@broadinstitute.org (J.G.T.Z.); Nikhil_Wagle@DFCI.HARVARD.EDU (N.W.); 3Department of Biology, The Pennsylvania State University, University Park, PA 16802-6300, USA; rza1@psu.edu; 4Department of Medical Oncology, Dana-Farber Cancer Institute, Harvard Medical School, Boston, MA 02115, USA; Anthony_Letai@dfci.harvard.edu; 5Human Oncology & Pathogenesis Program, Memorial Sloan Kettering Cancer Center, New York, NY 10065, USA; maurizio.scaltriti@astrazeneca.com; 6Department of Electronics and Biomedical Engineering, University of Barcelona (UB), 08028 Barcelona, Spain; 7Networking Biomedical Research Center in Bioengineering, Biomaterials and Nanomedicine (CIBER-BBN), 28029 Madrid, Spain

**Keywords:** ER+ breast cancer, DBP, priming, apoptosis, targeted therapies, resistance, BH3 mimetics

## Abstract

Breast cancer is the most frequent type of cancer and the major cause of mortality in women. The rapid development of various therapeutic options has led to the improvement of treatment outcomes; nevertheless, one-third of estrogen receptor (ER)-positive patients relapse due to cancer cell acquired resistance. Here, we use dynamic BH3 profiling (DBP), a functional predictive assay that measures net changes in apoptotic priming, to find new effective treatments for ER+ breast cancer. We observed anti-apoptotic adaptations upon treatment that pointed to metronomic therapeutic combinations to enhance cytotoxicity and avoid resistance. Indeed, we found that the anti-apoptotic proteins BCL-xL and MCL-1 are crucial for ER+ breast cancer cells resistance to therapy, as they exert a dual inhibition of the pro-apoptotic protein BIM and compensate for each other. In addition, we identified the AKT inhibitor ipatasertib and two BH3 mimetics targeting these anti-apoptotic proteins, S63845 and A-1331852, as new potential therapies for this type of cancer. Therefore, we postulate the sequential inhibition of both proteins using BH3 mimetics as a new treatment option for refractory and relapsed ER+ breast cancer tumors.

## 1. Introduction

Breast cancer is the most common type of cancer in women and one of the top causes of mortality worldwide [1,2]. Enormous efforts have been devoted to studying this complex and heterogeneous disease, classifying it by molecular signature, predicting response to therapies, and improving patient prognosis. It is now well established that there are several main breast cancer subtypes: estrogen receptor (ER) positive and/or progesterone receptor positive; human epidermal receptor 2 (HER2) amplified; and a third group that does not express any of the receptors above, referred as triple-negative breast cancer [2]. This acquired knowledge in breast cancer biology led to the identification of novel molecular targets and the development of specific inhibitors against them [3]. Recently, newly developed compounds targeting key altered proteins, such as ER, EGFR, HER2, MET, PI3K, AKT, mTOR, MAPK pathway, PARP, and many others, induce cancer cell elimination [4,5,6,7]. Some of these agents are currently used in the clinic, such as ER inhibitors, lapatinib, and trastuzumab against HER2, or PARP inhibitors such as olaparib [8,9,10,11,12,13], and others.

Three-quarters of all breast cancers are ER positive (ER+) and present signaling deregulation. ER is a protein with a ligand and a DNA-binding domain that, once activated, acts as a transcription factor regulating gene expression, cell proliferation, development, and multiple physiological roles [14]. Generating specific drugs against this receptor has been an extensive field of study to treat ER+ tumors for decades. In this regard, one of the first developed anti-estrogens was tamoxifen, which blocks ER activity [15], achieving outstanding clinical results and overall improving breast cancer patient survival [14]. Other compounds employing different mechanisms of action followed, such as fulvestrant (that induces ER destabilization and degradation [15]) or aromatase inhibitors that reduce estrogen production [15], similarly ameliorating patient treatment [16]. In fact, most women respond to these ER-targeting agents and survive the disease, yet one-third of them relapse, presenting a high risk of metastasis [17]. A myriad of mechanisms of resistance have been described by different laboratories, including ESR1 mutations, compensation by other receptors (such as androgen and progesterone receptors), changes in expression/activation of ER-signaling proteins, drug metabolism, endocrine secretion, and others [14]. Therefore, there is a clear unmet need for new ER+ breast tumor treatments after tamoxifen or aromatase inhibitor-based therapy recurrence [14].

Most anticancer agents used to eliminate breast tumors kill through apoptotic programmed cell death [4,18,19,20,21,22]. Apoptosis is a form of cell death that is regulated by the BCL-2 family of proteins that controls mitochondrial outer membrane permeabilization, the point of no return for this process [23]. When a cancer cell is effectively treated, early changes in the BCL-2 family of proteins can be detected that precede apoptotic process engagement [24,25]. In this regard, functional assay dynamic BH3 profiling (DBP) can measure these initial pre-apoptotic events and predict later cytotoxicity. In fact, it has been proven as an excellent predictive biomarker for anticancer therapy response to a wide variety of targeted agents in breast cancer cell lines and in primary patient samples [26,27]. Moreover, DBP can be used to identify anti-apoptotic changes upon treatment using specific synthetic peptides to determine increased dependence on BCL-2, BCL-xL, or MCL-1. Using this information, we can explore combinations with BH3 mimetics, such as ABT-199 (also known as venetoclax, a BCL-2 inhibitor) [28], S63845 (MCL-1 inhibitor) [29], or A-1331852 (A-133) (BCL-xL inhibitor) [30], that block anti-apoptotic proteins to restore cell death [31]. Most of these molecules are now being evaluated in clinical trials as single agents or especially in combination to treat different forms of cancer [32]. Notably, several laboratories have demonstrated the potential use of BH3 mimetics to improve HER2+ [33], PI3KCA mutant [34] and ER+ [35] breast cancer treatment. However, how breast cancer cells adapt and survive therapy leading to disease progression in the clinic it is still poorly understood.

Here, we use DBP to identify ER+ breast cancer anti-apoptotic adaptations to survive therapy. We demonstrate that when blocking these survival mechanisms with specific BH3 mimetics, cancer cells rapidly adapt and use a second anti-apoptotic protein. By timely identification of this compensation, we can sequentially block these anti-apoptotic proteins with BH3 mimetics to overcome tumor adaptation to treatment, reaching high cytotoxicity in ER+ cancer cells where other standard therapies fail.

## 2. Materials and Methods

### 2.1. Cell Lines and Treatments

Breast cancer cell lines MDA-MB-415, T47D, and MCF7 were purchased from American Type Culture Collection (ATCC, Manassas, Virginia). Cells were tested for mycoplasma and cultured in RPMI 1640 medium (31870, Thermo Fisher, Gibco, Paisley, Scotland) supplemented with 10% heat inactivated fetal bovine serum (10270, Thermo Fisher, Gibco, Paisley, Scotland), 1% L-glutamine (25030, Thermo Fisher, Gibco, Paisley, Scotland) and 1% penicillin and streptomycin (15140, Thermo Fisher, Gibco, Paisley, Scotland) and maintained at 37 °C in a humidified atmosphere of 5% CO_2_. Drug treatments were performed directly in the culture media at the doses and time points indicated in every single experiment. All drugs were purchased at Selleckchem (Munich, Germany).

### 2.2. Dynamic BH3 Profiling

Dynamic BH3 profiling experiments were performed as previously described [36,37]. In brief, 3 × 10^4^ cells/well in a 96-well plate were used for cell lines. A 25 μL aliquot of BIM BH3 peptide (final concentration of 0.01, 0.03, 0.1, 0.3, 1, 3, and 10 μM), 25 μL of BAD BH3 peptide (final concentration of 10 μM), 25 μL of HRK BH3 peptide (final concentration of 100 μM), and 25 μL of MS1 BH3 peptide [38] (final concentration of 10 μM) in MEB (150 mM mannitol, 10 mM Hepes-KOH pH 7.5, 150 mM KCl, 1 mM EGTA, 1 mM EDTA, 0.1% BSA, 5 mM succinate) with 0.002% digitonin were deposited per well in a 96-well plate (3795, Corning, Madrid, Spain). Single-cell suspensions were stained with the viability marker Zombie Violet (423113, BioLegend, Koblenz, Germany) and then washed with PBS and resuspended in MEB in a final volume of 25 μL. Cell suspensions were incubated with the peptides for 1 h following fixation with formaldehyde and staining with cytochrome c antibody (Alexa Fluor^®^ 647 anti-cytochrome c-6H2.B4, 612310, BioLegend, Koblenz, Germany). Individual DBP analysis were performed using triplicates for DMSO, alamethecin (BML-A150-0005, Enzo Life Sciences, Lorrach, Germany), the different BIM BH3 concentrations used, and BAD, HRK, and MS1 BH3 peptides. The expressed values stand for the average of three different readings performed with a high-throughput flow cytometry SONY instrument (SONY SA3800, Surrey, United Kingdom). Δ% priming stands for the maximum difference between treated cells minus non-treated cells for a given peptide.

### 2.3. Cell Death Analysis

Cells were stained with fluorescent conjugates of Annexin V (FITC Annexin V, 640906 or Alexa Fluor^®^ 647 Annexin V, 640912, BioLegend, Koblenz, Germany) and propidium iodide (PI) (1056, BioVision, Milpitas, CA, USA) and analyzed on a flow cytometry Gallios instrument (Beckman Coulter, Nyon, Switzerland). Viable cells are Annexin V negative and PI, and cell death is expressed as 100% viable cells.

### 2.4. Protein Extraction and Quantification

Proteins were extracted by lysing the cells during 30 min at 4 °C using RIPA buffer (150 mM NaCl, 5 mM EDTA, 50 mM Tris-HCl pH = 8, 1% Triton X-100, 0.1% SDS, EDTA-free Protease Inhibitor Cocktail (4693159001, Roche, Mannkin, Germany)) followed by a centrifugation at 16,100*g* for 10 min. The supernatant was stored at −20 °C for protein quantification performed using Pierce ^TM^ BCA Protein Assay Kit (23227, Thermo Fisher, Paisley, Scotland).

### 2.5. Immunoprecipitation

Cells were lysed in immunoprecipitation buffer (150 mM NaCl, 10 mM Hepes, 2 mM EDTA, 1% Triton, 1.5 mM MgCl2, 10% glycerol and EDTA-free Protease Inhibitor Cocktail (4693159001, Roche, Mannkin, Germany)) and centrifuged at 14,000× *g*, 15 min at 4 °C. The resulting supernatants were incubated with magnetic beads (161-4021, Bio-Rad, Madrid, Spain) conjugated to 5 µg of rabbit anti-BIM antibody (CST2933, Cell Signaling, Leiden, The Netherlands) or 5 μg of rabbit IgG antibody (CST2729, Cell Signaling, Leiden, The Netherlands) at 4 °C overnight. A fraction of the supernatant (30 μL) was removed and mixed with half volume of 4× SDS-PAGE sample buffer, heated at 96 °C for 5 min, and stored at −80 °C as cell lysate fractions. After magnetization, a part of the supernatant was mixed with half volume of 4× SDS-PAGE sample buffer, heated at 96 °C for 5 min, and stored at −80 °C as unbound fractions. The rest of the supernatant was discarded. The resulting pellet was washed and mixed with 40 µL 4× SDS-PAGE sample buffer and heated 10 min at 70 °C to allow dissociation between the purified target proteins and the bead–antibody complex. Finally, sample was magnetized, and the supernatant was collected and stored at −80 °C as immunoprecipitation (IP) fractions for further Western blot analysis.

### 2.6. Immunoblotting

Proteins were separated by SDS-PAGE (Mini-Protean TGX Precast Gel 12%, 456-1045, Bio-Rad, Madrid, Spain) and transferred to PVDF membranes (10600023, Amersham Hybond, Pittsburgh, PA, USA). Membranes were blocked with dry milk dissolved in Tris-buffered saline with 1% Tween 20 (TBST) for 1 h and probed overnight at 4 °C with the primary antibodies of interest directed against rabbit anti-BCL-xL (CST2764, Cell Signaling, Leiden, The Netherlands), rabbit anti-MCL-1 (CST5453, Cell Signaling, Leiden, The Netherlands), rabbit anti-BIM (CST2933, Cell Signaling, Leiden, The Netherlands), rabbit anti-actin (CST4970, Cell Signaling, Leiden, The Netherlands) followed by anti-rabbit IgG HRP-linked secondary antibody (CST7074, Cell Signaling, Leiden, The Netherlands) in 3% BSA in TBST for 1 h at room temperature. Immunoblots were developed using Clarity ECL Western substrate (1705060, Bio-Rad, Madrid, Spain). When necessary, immunoblots were stripped in 0.1 M glycine pH 2.5, 2% SDS for 40 min, and washed in TBS. Bands were visualized with LAS4000 imager (GE Healthcare Bio-Sciences AB, Uppsala, Sweden) and ImageJ were then used to measure the integrated optical density of bands.

### 2.7. Statistical Analysis

Statistical significance of the results was analyzed using Student’s t-tail test. * *p* < 0.05 and ** *p* < 0.01 were considered significant. SEM stands for standard error of the mean. For ROC curve analysis, cell lines were considered responsive to treatment when Δ% cell death > 25%. Drug synergies were established based on the Bliss Independent model as previously described [39]. Combinatorial index (CI) was calculated CI = ((D_A_ + D_B_) − (D_A_ × D_B_))/D_AB_, where D represents cell death of compound A or B or the combination of both. Only the combination of drugs with a CI < 1 were considered synergies. GraphPad Prism8 was used to generate the graphs and to perform the statistical analysis.

## 3. Results

### 3.1. Dynamic BH3 Profiling Predicts Targeted Agents’ Effectiveness in ER+ Breast Cancer Cells

Breast cancer survival has increased in recent decades, partially due to the introduction of targeted therapies [3]. Beyond drugs that directly target proteins such as ER, EGFR, HER2, MET, PI3K, AKT, mTOR, and those of the MAPK pathway [7], anti-apoptotic protein inhibitors (BH3 mimetics) are now being evaluated as new targeted therapies for breast cancer [35] as increased anti-apoptotic protein levels have been reported [33,35,40]. Nevertheless, as one-third of ER+ breast cancer patients relapse following current therapies and present a high risk of metastasis [17], we sought to identify new strategies to better treat this type of cancer. We selected different targeted agents that are currently evaluated in pre-clinical and clinical trials. We used ipatasertib (AKT inhibitor) [41], which has already been tested for triple-negative breast cancer [42]; everolimus (mTOR inhibitor), alpelisib (PI3K inhibitor) [43,44], palbociclib (CDK4/6 inhibitor) [45], fulvestrant (estrogen receptor antagonist) [46,47], that have been approved for hormone receptor positive breast cancer treatment, and the BH3 mimetics S63845 (MCL-1 inhibitor) [33] and ABT-199 (BCL-2 inhibitor) [35], which are currently being explored in clinical trials. Using DBP, we first tested the overall response to treatments using the BIM peptide [26] in two ER+ cell lines, MDA-MB-415 and T47D [48], after 16 h of incubation with the described drugs. MDA-MB-415 cells showed an increase in Δ% priming after ipatasertib and S63845 (Figure 1A); in contrast, alpelisib, ABT-199, everolimus, palbociclib, and fulvestrant treatments did not produce any effect on these cells (Figure 1A). Interestingly, when we compared these DBP predictions with the concomitant cell death assessed by Annexin V and PI staining at 72 h, we observed a good correlation between the increase in Δ% priming with DBP (Figure 1A) and the increase in Δ% cell death (Figure 1B). T47D cells showed almost no increase in Δ% priming after the treatments mentioned before (Figure 1C). When we analyzed cytotoxicity at 72 h, we could just observe a modest significant increase in cell death after alpelisib treatment (Figure 1D). We also tested MCF7 cells by DBP and we could not observe any increase in the Δ% priming or significant cytotoxicity with any of the compounds analyzed (data not shown). We obtained a significant correlation between Δ% priming and Δ% cell death in the breast cancer cells analyzed (Figure 1E). To determine how good DBP is as a binary predictor for the targeted agents’ efficacy in breast cancer, we performed receiver operating characteristic (ROC) curve analysis [49]. The area under the curve (AUC) for a random classifier would be 0.5, whereas for a perfect predictor the AUC would be 1. Our results showed that the AUC was 1 (Figure 1F), indicating the excellent predictive capacity of DBP for the targeted agents and breast cancer cell lines tested. Collectively, these results showed the identification of effective targeted therapies for ER+ breast cancer cell lines using DBP.

### 3.2. Inhibition of Anti-Apoptotic Adaptations Could Overcome Treatment-Induced Resistance

Despite the development of different targeted therapies to treat ER+ breast cancer patients, such as tamoxifen, fulvestrant, or aromatase inhibitors [14,17], 15–20% of patients still relapse within 5 years of treatment withdrawal [50]. It has been previously described that the over-expression or over-activation of anti-apoptotic proteins can lead to disease progression [51]. In particular, it has been reported that tamoxifen-treated ER+ breast cancers often present high BCL-2 and BCL-xL expression [52], and that tumors harboring elevated MCL-1 protein expression exert poorer prognosis [53]. In this regard, DBP can anticipate anti-apoptotic adaptations to treatments [31] and guide the use of BH3 mimetics to overcome them. We previously identified some targeted therapies that caused cytotoxicity in ER+ cell lines (Figure 1) but we wanted to further investigate possible adaptations to those compounds and new strategies to increase their efficacy. Therefore, we analyzed the contribution for each anti-apoptotic protein using DBP by measuring the increase in Δ% priming after treatments using the BAD, HRK, and MS1 peptides that are specific for these proteins (Figure 2). We observed an increase in Δ% priming after ipatasertib with the HRK peptide in MDA-MB-415 but not in T47D cells (Figure 2A) indicating the BCL-xL dependence of the first. These results were further corroborated by the observation of a synergistic combination (CI < 1) [39] with the sequential treatment of ipatasertib followed by A-133 in MDA-MB-415 (CI = 0.588) (Figure 2B) but not in T47D cells (CI = 1.02) (Figure 2C). Moreover, we could also detect a strong adaptation through BCL-xL after S63845 treatment in both cell lines (Figure 2A), which was further confirmed by cell death analyses, observing a synergistic effect (CI = 0.362 for MDA-MB-415 and CI = 0.112 for T47D) when sequentially combining S63845 and A-133 in both cell lines (Figure 2B,C). Interestingly, when we did the opposite and treated MDA-MB-415 and T47D with A-133, we detected an increase in Δ% priming with the MS1 peptide (Figure 2D), indicating a reciprocal adaptation through MCL-1. Indeed, we observed again a synergistic combination (CI = 0.607 for MDA-MB-415 and CI = 0.105 for T47D) with the sequential treatment with A-133 followed by S63845 (Figure 2E,F), overcoming the predicted resistance. Thus, MCL-1 and BCL-xL clearly compensated for each other in ER+ breast cancer cells as the inhibition of one leads to cells escaping apoptosis through the other. Despite not obtaining any cytotoxic response with the single agents in MCF7 cells, we explored if these cells could develop similar anti-apoptotic adaptations, but we could not observe an increase in Δ% priming with the MS1 peptide after the treatment with A-133 nor an increase in Δ% cell death after the sequential treatment of S63845 and A-133 (data not shown). We hypothesize that MCF7 resistance to cell death could be partially explained by the caspase-3 deficiency that these cells present [54].

Finally, we also sought to study BCL-2 contribution to MCL-1 inhibition resistance (Appendix A). We observed a BCL-2 mediated adaption to S63845 treatment as an increase in Δ% priming was observed with the BAD peptide after 16 h of treatment in MDA-MB-415 and T47D cell lines (Appendix A). Cell death analyses demonstrated a synergistic effect when sequentially combining both drugs in MDA-MB-415 (CI = 0.773) and T47D cells (CI = 0.820) (Appendix A). Nevertheless, the contribution of BCL-2 to MCL-1 inhibition is minor compared to the one observed with BCL-xL, indicating that the anti-apoptotic MCL-1/BCL-xL axis is the predominant one in breast cancer, as previously reported [55]. These results exemplify that DBP can accurately predict anti-apoptotic adaptations to treatments after only 16 h of incubation and can be used to design effective rational sequential combinations of targeted agents. In summary, the most effective metronomic combinations that we identified in ER+ breast cancer cell lines were from concomitantly administering the BH3 mimetics S63845 and A-133.

### 3.3. Resistance to Treatments Relies on BCL-xL and MCL-1 Binding to BIM

To better understand the molecular mechanism underlying the observed anti-apoptotic adaptations, we decided to explore the expression of anti-apoptotic proteins after treatment in T47D cells. We focused on MCL-1 and BCL-xL after treatment with the BH3 mimetics S63845, ABT-199, and A-133 (Figure 3). Surprisingly, we could not detect neither an increase in BCL-xL expression after S63845 treatment nor an increase in MCL-1 after A-133 treatment (Figure 3), as could be expected. These results point to a different molecular explanation rather than an anti-apoptotic protein overexpression to rescue ER+ breast cancer cells from death. As anticipated, we could observe an increase in MCL-1 protein expression after S63845 treatment due to an extension of the protein half-life as has been previously reported [56].

As the total amount of anti-apoptotic proteins (Figure 3) could not explain the observed adaptations, and as it has been shown that MCL-1 and BCL-xL share the ability to bind with high affinity to BIM [57,58], we wondered if their interaction could confer the treatment resistance. To answer this question, we performed immunoprecipitation assays (IP). We first analyzed the IP efficiency by checking BIM expression in the unbound fraction (Figure 4A), and we then confirmed that no relevant changes in total protein levels were detected after treating the cells (Figure 4B). When we analyzed MCL-1 and BCL-xL bound to BIM after the selected treatments, we observed that A-133 induced a significant decrease in BCL-xL binding to BIM and a significant increase in MCL-1 interaction with BIM (Figure 4C). Similarly, S63845 treatment induced a significant decrease in the amount of MCL-1 bound to BIM while it significantly increased with BCL-xL (Figure 4C). These results demonstrate that these anti-apoptotic proteins have redundant functions and that they compensate one another to avoid apoptotic cell death in ER+ breast cancer cells (Figure 5). Thus, when we combined both BH3 mimetics sequentially, BIM could no longer bind to MCL-1 or BCL-xL, triggering BAX and BAK activation and restoring apoptotic cell death (Figure 5).

## 4. Discussion

Breast cancer is the major cause of mortality in women [1]. Despite the identification of different molecular targets and the development of specific inhibitors against them [3], resistance toward these treatments leads to disease progression. Particularly in ER+ breast cancers, one-third of patients relapse and present risk of metastasis after treatment with ER-targeting agents [17]. Here, we used DBP to identify new potential treatments with BH3 mimetics for ER+ breast cancer. In fact, we used DBP to predict the efficacy of ipatasertib and S63845 in MDA-MB-415 cells which correlated with later cytotoxicity (Figure 1A,B). Despite observing cytotoxicity when treating with ipatasertib, we could not detect it with alpelisib, as it was expected due to the loss of function mutation in PTEN occurring in this cell line [59,60] (Figure 1B). Similar results have been reported before and could be linked to different regulation steps of the signaling pathway, such as the distinct levels of PIK3CA pathway activation, the AKT signaling independent of PIK3CA, the crosstalk with other pathways, and the complex feedback regulation of the pathway itself [61]. This level of complexity has been previously reported in a detailed network model that explored the resistance mechanisms to PI3K inhibitors [62]. Therefore, we postulate the possible use of an AKT inhibitor to treat ER+ breast cancer similarly to what has been reported for metastatic triple-negative breast cancer [63]. Furthermore, the high cytotoxicity showed by the BH3 mimetic S63845 reinforces the idea that one of the key anti-apoptotic proteins in breast cancer is MCL-1 [49]. In T47D cells, we observed a much lower response to all single agents tested, with the sole exception of alpelisib (Figure 1D) as these cells harbor activating PIK3CA mutation [60]. However, we could not observe cytotoxic effects of ipatasertib (Figure 1D), which can be linked to the weak phosphorylation of Akt observed in these cells despite the described activating mutation and gain of copy number of PIK3CA [64]. These results also showed the different responses to treatments that cell lines from the same cancer subtype could display. For this reason, the use of DBP could be crucial for the identification of new effective treatments for different ER+ cancer patients that are no longer responding to endocrine therapy.

One of the hallmarks of cancer is cell death evasion and resistance to therapy, and a rapid mechanism to mediate this is through anti-apoptotic BCL-2 family proteins [52]. It has been recently reported that resistance to MCL-1 treatment in a metastatic model of breast cancer could be mediated by BCL-xL [49]; in fact, we validated this observation performing DBP with the HRK peptide (Figure 2A), as we observed a synergistic combination when sequentially combining S63845 and A-133 (Figure 2B). However, the contribution of the anti-apoptotic protein BCL-2 is clearly minor (Appendix A), confirming the importance of BCL-xL in mediating the resistance to MCL-1 inhibition in breast cancer, as previously described [48]. Similarly, we also observed that when blocking BCL-xL with A-133, T47D cells survive through MCL-1 (Figure 2C,D), highlighting the importance of these two anti-apoptotic proteins in BH3 mimetics adaptation in breast cancer. One could expect that these combinations of treatments would have the same cytotoxic effect when applied together, however it has been previously reported that the sequential combination—but not the simultaneous coadministration—of targeted therapies and chemotherapeutic agents dramatically sensitizes breast cancer cells and improve therapeutic efficacy [65]. Furthermore, the sequential administration of drugs would decrease the toxicity in patients while having the same cytotoxic effect, thus reinforcing the sequential treatment strategy presented here.

When we explored the molecular mechanism underlying these anti-apoptotic resistances, we found that the pro-apoptotic protein BIM shifts from one anti-apoptotic protein to the other depending on the BH3 mimetic used (Figure 4). BIM is therefore sequestered by MCL-1 when BCL-xL is inhibited with A-133, and by BCL-xL when MCL-1 is blocked with S63845, mediating rapid resistance to treatments to avoid apoptosis (Figure 5). Importantly, when we sequentially combine both BH3 mimetics, we displace BIM from these two anti-apoptotic proteins to activate BAX and BAK and engage apoptosis (Figure 5). We demonstrate here the crucial role of MCL-1 and BCL-xL in ER+ breast cancer, reinforcing the possible therapeutic use of BH3 mimetic combinations for this type of cancer to avoid patient relapse.

## Figures and Tables

**Figure 1 cells-10-01659-f001:**
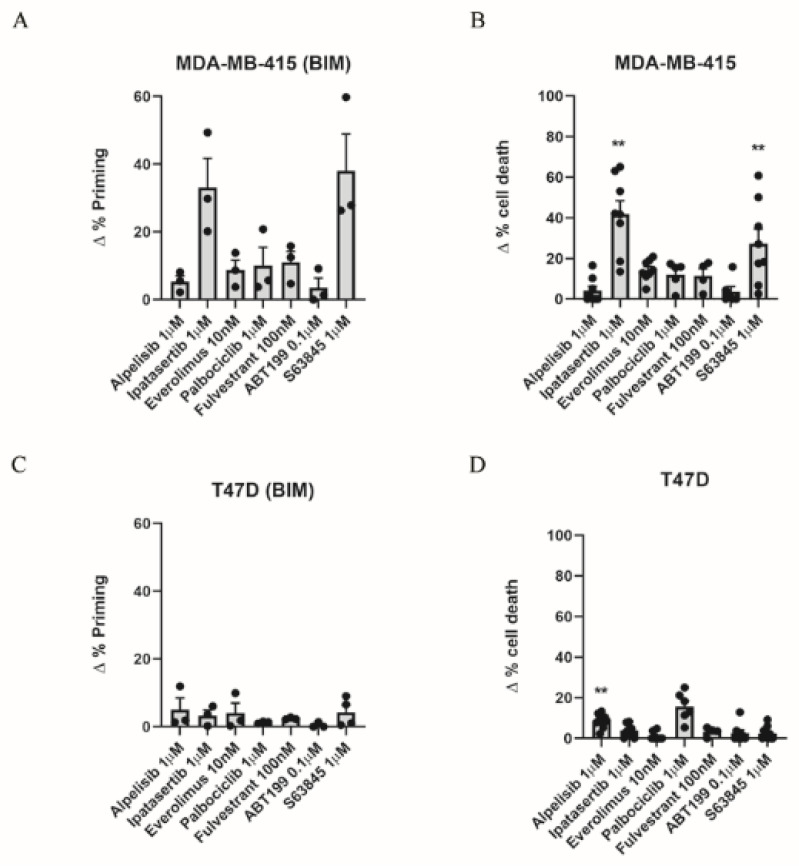
Dynamic BH3 profiling predicts sensitivity to targeted agents in different ER+ breast cancer cell lines. (**A**) Results from the DBP assay after 16 h incubation with different treatments in MDA-MB-415 cells. Results expressed as ∆% priming represents the increase in priming compared to control cells. (**B**) Cell death results from Annexin V and propidium iodide staining and FACS analysis after 72 h incubation with the targeted agents in MDA-MB-415 cells. Results expressed as ∆% cell death represents the increase in cell death compared to control cells. (**C**) Results from the DBP assay after 16 h incubation with different treatments in T47D cells. Results expressed as ∆% priming represents the increase in priming compared to control cells. (**D**) Cell death results from Annexin V and propidium iodide staining and FACS analysis after 72 h incubation with the targeted agents in T47D cells. Results expressed as ∆% cell death represents the increase in cell death compared to control cells. (**E**) Correlation analysis between Δ% priming and Δ% cell death in MDA-MB-415 and T47D cells. (**F**) Receiver operating characteristic curve analysis. Values indicate mean values ± SEM from at least three independent experiments. ** *p* < 0.01.

**Figure 2 cells-10-01659-f002:**
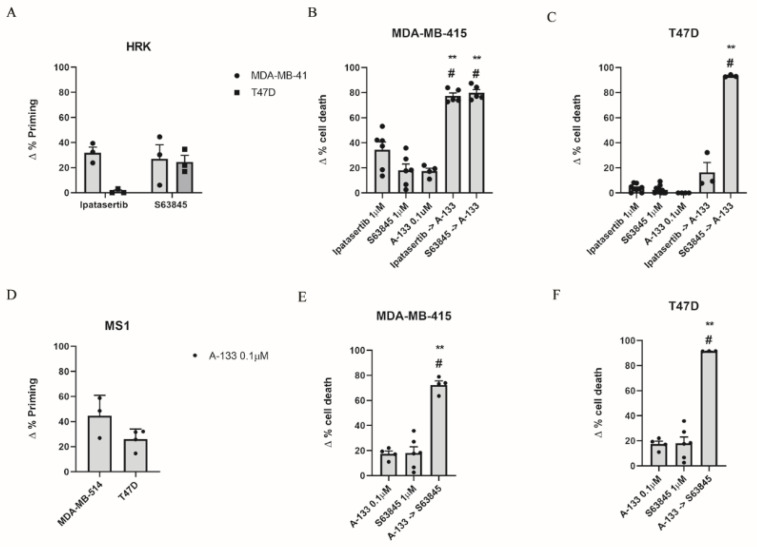
Dynamic BH3 profiling predicts BCL-xL and MCL-1 anti-apoptotic adaptation as a resistance mechanism after targeted agent treatment in ER+ breast cancer cell lines. (**A**) Results from the contribution of BCL-xL anti-apoptotic protein using the HRK peptide after ipatasertib 1 µM and S63845 1 µM treatment in MDA-MB-415 and T47D. Results expressed as ∆% priming represents the increase in priming compared to control cells. (**B**,**C**) Cell death from Annexin V and propidium iodide staining and FACS analysis after 72 h incubation of MDA-MB-415 and T47D cells with the single agents alone or the sequential combination of ipatasertib or S63845 with A-133. (**D**) DBP from the contribution of MCL-1 anti-apoptotic protein using the MS1 peptide after A-133 0.1 µM treatment. (**E**,**F**) Cell death analysis after 72 h incubation of MDA-MB-415 and T47D cells with the single agents alone or the sequential combination of A-133 and S63845 for 72 h. Values indicate mean values ± SEM. ** *p* < 0.01 compared to single agents and # indicates CI < 1. All experiments were performed at least three times.

**Figure 3 cells-10-01659-f003:**
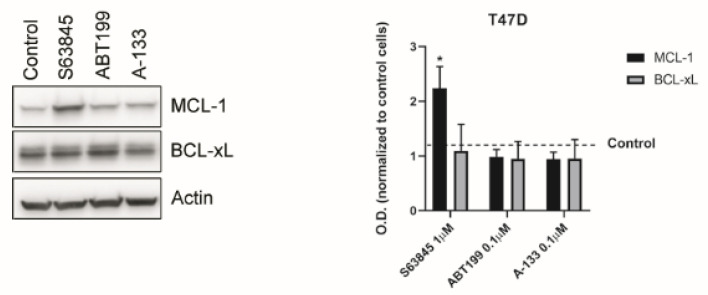
The identified resistant mechanisms are not due to overexpression of anti-apoptotic proteins. Left panel: Representative images from Western blot analysis of T47D control lysates and after treatment with BH3 mimetics for 16 h. Right panel: Optical density quantification normalized to actin and represented as fold change compared to control. S63845 treatment significantly increases MCL-1 expression due to its stabilization. Values indicate mean values ± SEM. * *p* < 0.05 and all experiments were performed at least three times.

**Figure 4 cells-10-01659-f004:**
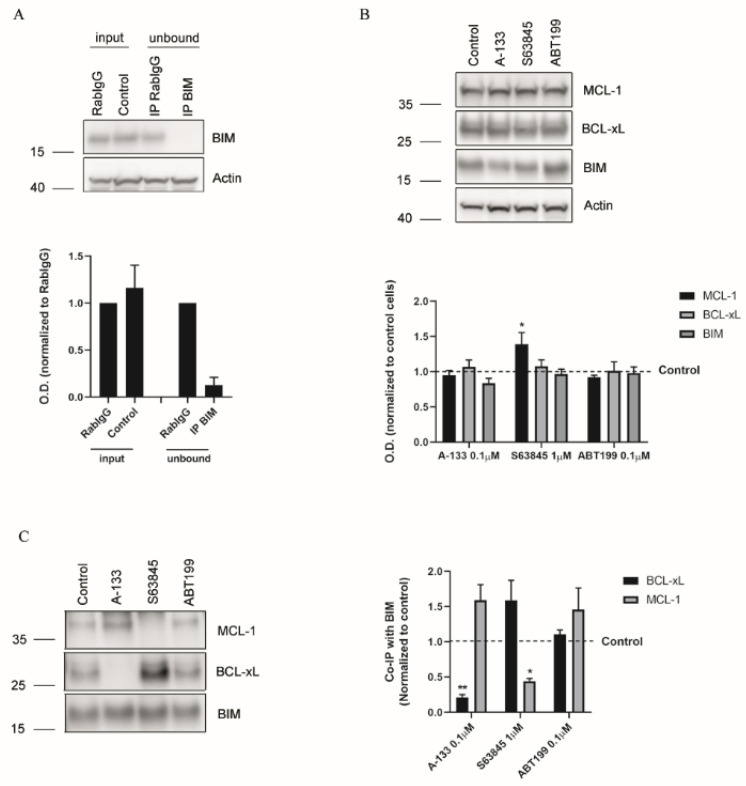
The acquired resistance mechanisms are controlled by the amount of anti-apoptotic protein bound to BIM. (**A**) Upper panel: Representative images of Western blot analysis of T47D cell lysates and unbound fractions after BIM immunoprecipitation. Lower panel: Optical density quantification of BIM normalized with actin levels and represented as fold increase compared to the control condition RabIgG. (**B**) Upper panel: Representative images of Western blot analysis of T47D cell lysates after 16 h treatment with the indicated drugs. Lower panel: Optical density quantification of each protein normalized to actin and represented as fold increase compared to control cells (**C**) Left panel: Representative images of Western blot analysis of BIM immunoprecipitation in T47D cells. Right panel: Quantification of the optical density of each protein and represented as binding ratio between BIM and MCL-1 or BCL-xL. Results expressed as fold increase represents the increase in optical density after treatments compared to control cells. Values indicate mean values ± SEM. ** *p* < 0.01, * *p* < 0.05 and all experiments were performed at least three times.

**Figure 5 cells-10-01659-f005:**
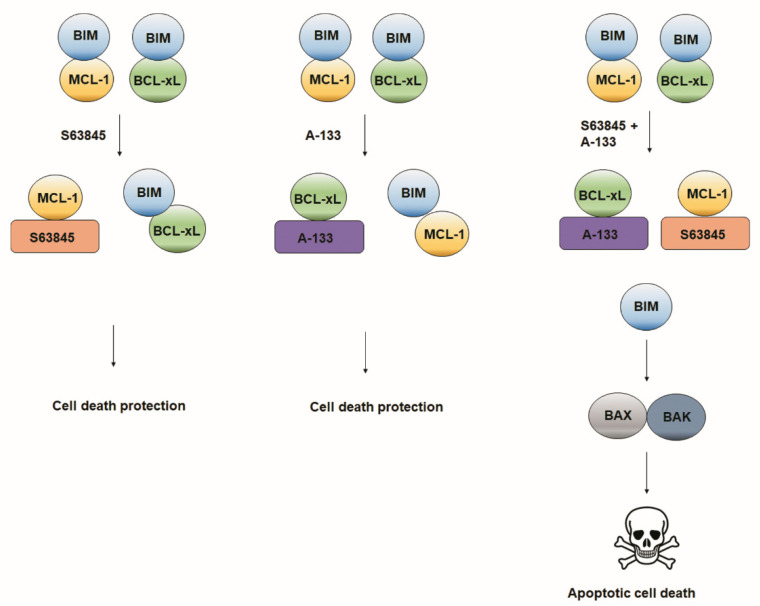
Schematic representation of MCL-1 and BCL-xL interaction with BIM as a therapy-acquired resistance mechanism. The model distinguishing mechanisms that may operate in the presence of either S63845, A-133, or the sequential combination of both BH3 mimetics. The interaction of BIM with MCL-1 and A-133 would shift depending on the BH3 mimetic used, conferring cell death protection. Only when we sequentially combined both BH3 mimetics, cells will undergo apoptotic cell death.

## Data Availability

Not applicable.

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
