# Peer review of "ER+ Breast Cancer Strongly Depends on MCL-1 and BCL-xL Anti-Apoptotic Proteins"

_cells, 2021, doi:10.3390/cells10071659_

Round 1

Reviewer 1 Report

The authors have adapted DBP to characterize apoptosis resistance mechanisms of two well-known model breast cancer cell lines. The results demostrate compesating adaptation mechanism of anti-apoptotic proteins and underline the potential benefit of sequential tretment modalities with different drugs and the the ude os DBP to test diferent drugs with patient-isolated cancer cells

Overall the paper is interesting and the conclusions are supperted by the results. There is still one potentially major issue to be clarified in the study protocol:

Question: The cells were incubated with the BH3 prptides at the presence of 1 mM EDTA and EGTA, both strongly binding Ca2+. It is well-known that increase in intracellular calcium occurs during the early and late steps of apoptosis. Could the presented results have been affected by cellular calcium deprivation?

Minor issue

Figure 3: MCL-1 protein expression was significantly elevated upon S63845 treatment. Although brought up in the main paragraph text, this should be also indicated in the bar graph and figure legend. Also the legend hader then needs some modification to not so strongly underline the expression related mechanism

Author Response

We thank the reviewer for the positive assessment of our work. We have revised our manuscript and we believe that this new version is now improved.

We appreciate this question from the reviewer because it allows us to clarify the use of this technique. The primary issue in BH3 profiling is avoiding the MPTP (mitochondrial permeabilization transition pore) which can be triggered by calcium and is independent of the BCL-2 family. The BH3 profiling conditions are artificial, in the same way many other bioassays are. In this case, calcium chelation is used to preserve the behavior of the BCL-2 family and the mitochondria during this assay, and our buffers are optimized for this purpose. We use a buffer system that preserves our ability to measure the desired phenotype reliably (cytochrome c release) with minimal influence from potential off-target effects, like the release of other cell components (proteases, calcium) due to physical disruption of membranes.

We appreciate the reviewer’s suggestion to improve the quality of the figure and we addressed it in this new version of the manuscript (see attached file).

Reviewer 2 Report

The manuscript from Alcon et al is well written and the results are quite well described. But there is no novelty in their study or improvement in the field. I think that the authors have to add more significative data before to consider the article for publication on Cells.

Major concerns:

-the authors carried out their experiments by using only two lines of breast cancer. I strongly recommend to use others like MDA-MB-231, MCF7 and MCF-RAS transformed their claim would definitively improve.

-statistics data are presented in a very rude form. The histograms need to be presented as bars with dot blot that are indicative of the number of samples and biological repetition.

-blots need to be quantified

- the results must be supported by more direct scientific evidences.

Author Response

We thank the reviewer for the comments and suggestions. Having only 10 days to submit our revision, we made some substantial additions, and we believe that this new version is now improved.

We appreciate the reviewer’s suggestion to improve the quality of the paper. We performed DBP on MCF7 cell applying the same treatments that we used in T47D and MDAMB415 cell lines. We observed that none of these treatments significantly increase the Δ % Priming analyzed by DBP (see attached file). We then analyzed the cytotoxic effect of some of these compounds by Annexin V and propidium iodide staining, correlating with the DBP predictions, none of these treatments increase the Δ % cell death in MCF7 cells (see attached file). The resistance of these cells to these treatments could be explained by their caspase 3 deficiency previously reported for this cell line (Jänicke, R.U. MCF-7 breast carcinoma cells do not express caspase-3. Breast Cancer Res Treat 117, 219–221 (2009)). We finally decided not to include this data in the manuscript although we now mention it in the text.

Nevertheless, we tried to identify some anti-apoptotic adaptation after ipatasertib, S63845 and A-133 treatments, as we did with T47D and MDAMB415 cells, that could be overcame by the sequential addition of a BH3 mimetic. We could not identify any adaptation through BCL-xL, detected with the HRK peptide using DBP, after ipatasertib and S63845 treatments (see attached file). Similarly, we could not observe an increase in Δ % priming with the MS1 peptide after A-133 treatment in these cells (see attached file). As predicted by DBP, when we sequentially combine ipatasertib or S63845 with the BCL-xL inhibitor A-133, there was no increase in the % cell death analyzed by Annexin V and propidium iodide staining, or when we sequentially applied S63845 after A-133 treatment (see attached file). As previously mentioned, these cells are deficient in caspase-3 and the direct perturbation of the apoptotic pathway is not causing any cytotoxic effect. Therefore, we finally decided not to include it in the manuscript although we now mention it in the text.

We also appreciate the reviewer’s suggestion to improve the quality of the figures, so we changed the graphical representations and included a protein quantification.

We believe that in this new version we provide enough scientific evidence to prove our conclusions. We think that this work is novel because it uses a unique technology (dynamic BH3 profiling or DBP) to explore rational sequential combinations of BH3 mimetics to treat one of the most common types of cancer: ER+ breast cancer. BH3 mimetics are now extensively explored in preclinical research and clinical trials, also in breast cancer, but our approach is different than other efforts performed in this matter since we directly study how cancer cells rapidly modulate their anti-apoptotic machinery to escape cell death. In this sense, understanding the underlying pro-survival mechanism can help design better therapeutic strategies for this type of cancer. Moreover, by uniquely using sequential combinations guided by DBP we can reduce certain secondary effects, such as thrombocytopenia, while maximizing the therapeutic effect of these compounds in patients. To our knowledge, there are no studies exploring the sequential use of BCL-xL and MCL-1 inhibitors to treat ER+ breast cancer.    

Round 2

Reviewer 2 Report

I appreciate the effort of the authors. Now the manuscript is improved and suitable for plublication on Cells.